# The Role of Catestatin in Preeclampsia

**DOI:** 10.3390/ijms25052461

**Published:** 2024-02-20

**Authors:** Michalina Bralewska, Tadeusz Pietrucha, Agata Sakowicz

**Affiliations:** Department of Medical Biotechnology, Medical University of Lodz, Zeligowskiego 7/9, 90-752 Lodz, Poland; michalina.bralewska@umed.lodz.pl (M.B.); tadeusz.pietrucha@umed.lodz.pl (T.P.)

**Keywords:** catestatin, preeclampsia, pregnancy, chromogranin A

## Abstract

Preeclampsia (PE) is a unique pregnancy disorder affecting women across the world. It is characterized by the new onset of hypertension with coexisting end-organ damage. Although the disease has been known for centuries, its exact pathophysiology and, most importantly, its prevention remain elusive. The basis of its associated molecular changes has been attributed to the placenta and the hormones regulating its function. One such hormone is chromogranin A (CgA). In the placenta, CgA is cleaved to form a variety of biologically active peptides, including catestatin (CST), known inter alia for its vasodilatory effects. Recent studies indicate that the CST protein level is diminished both in patients with hypertension and those with PE. Therefore, the aim of the present paper is to review the most recent and most relevant in vitro, in vivo, and clinical studies to provide an overview of the proposed impact of CST on the molecular processes of PE and to consider the possibilities for future experiments in this area.

## 1. Introduction

The hypertensive disorders of pregnancy (HDP) include chronic hypertension, gestational hypertension (GH), preeclampsia/eclampsia, and chronic hypertension with superimposed preeclampsia. This group constitutes the second leading cause of maternal mortality worldwide, behind maternal hemorrhage [1], while hypertension itself represents the most common complication of pregnancy, affecting from 5% to 10% of all pregnancies [2]. The most recent diagnostic criteria define hypertension in pregnancy as a systolic blood pressure (SBP) greater than or equal to 140 mmHg and/or diastolic blood pressure (DBP) greater than or equal to 90 mmHg on two or more occasions, at least four hours apart.

Worldwide, preeclampsia (PE) and eclampsia alone account for more than 50,000 maternal deaths annually [3]. This pregnancy-specific disorder is characterized by the onset of hypertension together with end-organ damage. Although the clinical presentation of PE is well established, its etiology remains vague and the only known containment is premature delivery by Cesarean section. Furthermore, molecular studies are complicated by discrepancies between groups of patients caused by differences in the applied diagnostic criteria (Table 1). 

Although both PE and GH are among the HDP and the diagnosis criteria encompass the recognition of hypertension during pregnancy in previously normotensive women, the diseases are two distinct entities, as in PE, damage of one of the end-organs must be confirmed. Furthermore, the epidemiologic, pathogenetic, and hemodynamic characteristics are different for both of them. 

## 2. Pregnancy, the Sympathoadrenal Neuroendocrine System, and Its Association with Preeclampsia

The extensive systemic and local adaptations of the maternal cardiovascular system characterize each complicated and non-complicated gestation. These enable and support the growing metabolic demands of the fetus as well as allow for the delivery of oxygen and nutrients to the developing placenta. Among the systemic, hemodynamic adaptations, the most spectacular ones are as follows: (1) an increased cardiac output presenting the sharpest rise at the beginning of the gestation and the maximum peak at the early third trimester, and (2) the elevation of maternal blood volume being accompanied by a gradual reduction in the peripheral resistance of maternal vessels. The latter is possible thanks to the systemic vasodilatation of maternal vessels appearing before the end of placentation and completion of uteroplacental circulation, i.e., before week five of gestation. Interestingly, some of the maternal adaptations of the vascular system begin even before gestation, i.e., during the luteal phase of a menstrual cycle. At that time, the elevation of progesterone level is accompanied by an increase in the density of maternal functional β2 adrenergic receptors, although the general density of β2 adrenergic receptors is unaltered or even depleted in gestation, in comparison to non-pregnant women [6,7]. These receptors mediate the catecholamine, especially norepinephrine and epinephrine, response and play a significant role in the decrease in maternal vascular resistance. Indeed, the activation of β-adrenergic receptors causes the vasodilatation of the vascular bed and additionally leads to the relaxation of uterine muscles [8]. Moreover, the study on transgenic mice with the overexpression of β2 adrenergic receptors demonstrates that these receptors upregulate the cardiac output and heart rate [9]. The effect of the action of adrenergic receptors on the cardiovascular system was also observed in humans [10]. Therefore, it is no surprise that the sympathoadrenal neuroendocrine system is changed in pregnant women, in comparison to non-pregnant women. Interestingly, the uterine levels of norepinephrine, as well as dopamine, are significantly lower in normal pregnant patients when compared to non-pregnant women [11], as well as in comparison to hypertensive pregnant women [12]. Indeed, the preeclamptic patients present higher levels of studied catecholamines in uterine tissues with greater elevation in the placental bed and a higher level of norepinephrine in plasma in comparison to non-hypertensive expectant women [11]. Additionally, although the epinephrine level does not vary significantly between preeclamptic and normotensive pregnant women [12], the epinephrine/norepinephrine ratio is significantly elevated in women whose pregnancy is complicated by PE [11]. It is possible that the higher level of catecholamines in preeclamptic women is an effort of a maternal organism to stimulate the β2 adrenergic receptors. Indeed, the affinity state of β2 receptors the in maternal tissue of normotensive women is much higher than in the preeclamptic patients [6]. This depletion in the density of functional β2 receptors in preeclampsia seems to be related to the reduction in the total number of β2 adrenergic receptors in maternal tissues rather than the loss of their function [6]. This low-affinity state of adrenergic receptors may explain why glucose metabolism is dysregulated in preeclamptic women [13]. The β2 adrenergic receptors are involved in glucose uptake in both in vitro myotubes and the skeletal muscle of a healthy subject [14]. The ablation of β2 adrenergic receptors in mice results in hyperglycemia, but the supplementation of obese mice or the rat model for human type 2 diabetes mellitus by the β2 adrenergic agonists improves glucose tolerance in these rodents [14,15,16]. The state of reduction in the density of functional β2 receptors in preeclamptic women may have an additional consequence in maternal response to the other stimulators of these receptors, including the CgA and its derivative peptides such as CST [17]. Especially since the depletion of CST in the blood and placental bed of preeclamptic women was observed in numerous studies [18,19,20], CST seems to deepen this pathological state. It is possible that CST cooperates with the adrenergic receptors in the regulation of the generation of nitric oxide (NO), which is known for its vasodilatory effect on the maternal vessel [21]. Moreover, an in vitro study demonstrates that CST action through the β2 adrenergic receptors preserves the apoptotic process of cardiomyocytes in response to oxidative stress [22]. CST is also perceived as an endocrine/paracrine cardiac modulator; the inotropic effect of CST is mediated mainly by the β2 adrenergic receptors [23]. Interestingly, CST action by the adrenergic system also influences fat deposits as well as induces the lipolysis process and fatty acid oxidation. Fat cell function is regulated by the adrenergic system including the β2 adrenergic receptors, the activation of which stimulates the lipolysis process. The in vitro studies indicate that the high level of catecholamines observed in obese patients, as well as in preeclamptic women [12], leads to the β adrenergic and leptin receptors desensitization and to disturbances in the secretion of leptin. These processes are normalized by chronic CST treatment [24].

## 3. Abnormal Placentation and the Possible Pathogenesis of Preeclampsia

Each gestation begins with the placenta under hypoxic conditions (2–3% O_2_) and in a locally present inflammatory state. Such an environment favors the correct proliferation and differentiation of the trophoblastic cells, and supports matrix degradation and successful implantation [25]. Later on, maternal spiral arteries are remodeled and transformed into wide, low-resistance vessels supplying the developing fetus with oxygen and nutrients. Yet, in PE, this process is disrupted (Figure 1). The placenta cells undergo increased apoptosis and autophagy, which inhibit spiral artery remodeling and increase the release of placental debris, which enters the maternal circulation [26]. These processes, accompanied by alterations in the production of angiogenic factors, (e.g., sFlt1; soluble fms-like tyrosine kinase 1 or sEng; soluble endoglin, endothelin-1; ET-1), prolong hypoxia and inflammation [27,28]. In response to this poor perfusion and ischemia, the placenta starts to produce excessive amounts of proinflammatory factors (e.g., NfKB; nuclear factor kappa B or TNFα; tumor necrosis factor alpha), reactive oxygen species (ROS), and biologically active peptides, thus exacerbating maternal endothelium dysfunction and inducing vasoconstriction [28]. This ultimately results in the development of hypertension and proteinuria. PE is also characterized by the dysregulation of hormones such as the peptides chromogranin A (CgA) and its derivative catestatin (CST). Despite the fact that CgA itself was found in trophoblast in 1996, as of now it is not entirely clear what its role in pregnancy and its complications, like PE, may be. Thus, the presented review attempts to fill this existing gap and to present some new insights into the possible involvement of CgA-derived CST in the PE pathomechanism. 

## 4. Chromogranin A Derivative: Catestatin

Chromogranin A (CgA) is a 48-kDa acidic protein belonging to the granin family. Its main function is associated with the induction and promotion of secretory granule generation. Furthermore, CgA is commonly used as a marker in the diagnosis of neuroendocrine tumors [29]. However, more recent scientific research has been directed toward its influence on the course of pregnancy [20,30]. The enzymatic cleavage of CgA results in the production of a number of biologically active peptides, one of which is catestatin (CST; human CHGA_352–372_, bovine CHGA_344–364_, rat CHGA_367–387_), a 21-amino-acid-residue peptide named after its inhibitory effect on catecholamine (CA) release [31,32,33]. CST is also a potent inhibitor of neuropeptide Y, adenosine triphosphate, and CgA secretion, and has been found to manifest angiogenic and vasorelaxant activities [31,34,35]. 

The protein structure of CST includes a sequence [SSMKLSFRARAYGFRGPGPQL] that is highly conserved across various species and flanked by proteolytic cleavage sites [34]. However, the DNA sequence coding for the fragment of chromogranin A corresponding to catestatin is known to house various potential nucleotide variants that can influence the amino acid sequence. So far, five naturally occurring variants have been identified in the CST region of the human CgA genome: Gly364Ser (rs9658667), Pro370Leu (rs9658668), Arg374Gln (rs9658669) [36], Tyr363Tyr (rs9658666), and Gly367Val (rs200576557) [34,37]. Interestingly, the region coding amino acid 367 is home to an additional variant known as rs1280075530 (https://www.ncbi.nlm.nih.gov/snp; accessed on 10 January 2024), which replaces glycine with alanine or aspartic acid in position 367 of the chromogranin A chain; this corresponds to position 385 of the prechromogranin chain, as an 18-amino-acid fragment is removed from prechromogranin A to create chromogranin A. Interestingly, the substitution of a glycine residue by aspartic acid in position 367 of chromogranin A results in the human catestatin protein sharing 100% homology with the rat form (Figure 2) [38].

As the above polymorphisms are differentially distributed between ethnic groups, different populations may demonstrate altered CST activity. 

### 4.1. CST and Blood Pressure Regulation

The vasoconstriction and subsequent hypertension (HT) observed in PE have been attributed to a failure in maternal spiral artery remodeling, as well as alterations in the pool of angiogenic factors and excessive production of pro-inflammatory cytokines. The vasodilatory effect of CST was first reported by Kennedy et al. in 1998, who concluded that the hypotensive action of CST appears to be partially mediated by histamine release [39]. Since then, the role of CST in hypertension has been widely studied and confirmed by multiple in vitro, in vivo, and clinical studies. 

It was shown in vitro that CST binds to the α, β, δ, and γ subunits of nicotinic acetylcholine receptors (nAChR). As a noncompetitive agonist of nAChR, CST may play an autocrine regulatory role in neuroendocrine secretion [40]. Its action may be associated with the blockage of calcium ion (Ca^2+^) influx through voltage-gated calcium channels, which suppresses both CA release and *CHGA* transcription [41,42]. Significant down-regulation of α7nAChR activity has been observed in the decidual macrophages of PE women compared to healthy patients, with this being accompanied by a reduction in anti-inflammatory M2 macrophages and an increase in the proportion of pro-inflammatory M1 cells [43]. 

The downregulation of α7nAChR was found to be associated with the development of preeclampsia through increasing the release of pro-inflammatory cytokines and decreasing anti-inflammatory ones via the nuclear factor kappa B (NF-κB) pathway [44]. In addition, α7nAChR dysfunction was found to be connected with the progression of end-organ damage (EOD) in hypertension, which is also included in the PE diagnostic criteria [45]. Considering the agonist function of CST, it is worth exploring whether this peptide may influence α7nAChR activity.

According to O’Connor et al., CST is significantly diminished in hypertensive patients. Its decline in plasma seems to be a very early event (even pre-hypertensive), as indicated by the fact that the CST level appears to be lower in both hypertensive mothers and their normotensive offspring [46]. Hence, fluctuations in CST level may influence the pathophysiology of hypertension, rather than act as products. 

An in vivo study conducted on CST-knockout (CST-KO) mice confirmed that this peptide plays a role in blood pressure regulation, as the tested animals were hypertensive, with elevated serum cytokine levels and downregulated anti-inflammatory gene expression; their hearts also showed marked infiltration with macrophages [47]. All phenotypes were reversed by the exogenous administration of recombinant CST [48]. CST was also found to have a vasodilatory effect in a clinical study in which different CST concentrations were infused into the dorsal hand vein of normotensive men and women after pharmacologic venoconstriction with phenylephrine, resulting in vasodilation of the vessels [18]. The vasodilatory effect was more profound in women, as the study revealed that despite having lower levels of the CHGA precursor, female subjects displayed higher CST plasma protein levels. The reason may rest in the differences between the male and female endocrine responses, and the presence of gender-dependent CST genetic variants that may amplify those responses. 

One of the most important hormones responsible for developing female sexual characteristics is estrogen, which enacts its cellular responses by estrogen receptors (ERs) [49]. It was shown that estrogen negatively regulates the CST precursor CgA, resulting in a decrease in CgA mRNA [50]. In addition, as women exhibit higher CST protein levels than men, it is possible that the female organism has more efficient CgA cleavage mechanisms. The binding of estrogen to ER was shown to stimulate nitric oxide (NO) production and thus elicit vasodilatory effects triggered by the c-Src/ERK1/2/PI3K/Akt pathway (ERK1/2, extracellular signal-regulated kinases 1/2; PI3/K/Akt, phosphatidylinositol 3-kinase/ protein kinase B) [51]. 

CST is known for its vasodilatory action, which is associated with inter alia NO release [52]. It also performs various other functions that are mediated through c-Src/ERK1/2/PI3K/Akt pathway activation. Therefore, there is a possibility that CST may act through ER receptors. As estrogen function is associated mainly with the female reproductive organs, it is highly possible that either ERs are more numerous in the female organism or CST activity is higher than in men; as such, the vasodilatory effect of CST mediated by ER would be more profound in women, as it was observed in the in vivo study. 

The prevalence of CST polymorphisms appears to be strongly associated with ethnicity and sex and inhibits catecholamine secretion to differing degrees. For instance, the naturally occurring Gly364Ser variant may reduce or enhance the risk of hypertension, depending on the tested population [53,54]. The polymorphism has also been found to have a more profound protective effect in men. Further, the resequencing of *CHGA* exons revealed that the Pro370Leu CST variant increases catestatin activity and effectively inhibits nicotine-stimulated catecholamine secretion [37]. The Pro370Leu variant was found to be more resistant to plasmin digestion than Gly364Ser and Arg374Gln for generating a catestatin peptide fragment [55]. Similarly, Biswas et al. found that cathepsin L (CSTL) also failed to generate a CST peptide in this genetic variant [56]. As Pro370Leu effectively increases CST activity, it is clear that other proteolitic enzymes must be responsible for this peptide generation in the Pro370Leu CST genetic variant. 

A genome-wide association study for plasma catestatin conducted by Benyamin et al. confirmed early reports that CgA can be processed by kallikrein B plasma (Fletcher factor) 1 (KLKB1), resulting in the production of a biologically active CST peptide [57,58]. Further, the presence of the Ser124Asn genetic variant of KLKB1 correlated with elevated CST and lower CgA levels. 

A recommended goal for future studies would therefore be to examine the influence of different enzymes on each polymorphism with regard to population type. Such a holistic approach may shed light on the physiological pathways of CgA processing and yield new insights into the role of the CST in the development of human hypertension. 

Recent studies indicate higher *CHGA* gene expression in placentas obtained from PE women than in healthy controls, with a lower CST protein level in the study group [30]. Those outcomes confirm those of previous CST-hypertension-based studies [46,59,60], according to which CgA levels are increased in hypertensive patients, while CST protein level is diminished. However, some experiments and studies report high CST levels in hypertensive and PE patients [60,61]. This discrepancy, however, may result from the differences in the applied inclusion criteria, and the presence of coexisting diseases influencing CST production. Nevertheless, the majority of studies indicate that hypertensive or preeclamptic patients demonstrate lower basic CST levels than healthy patients. It is also possible that the CgA protein level is initially diminished, but increases as the disease progresses due to the upregulation of *CHGA* expression. However, this theory is directly dependent on a number of variables, with ethnicity, sex, genetic variants, and tissue of interest playing key roles, as well as the related proteolytic enzymes. Indeed, studies found both hypertensive patients and their offspring to exhibit lower CST levels [46]. A similar situation was observed in a previous in vitro study based on trophoblastic BeWo and HTR-8/SVneo cell lines, in which cells cultured in preeclamptic and early normotensive environments both demonstrated low CgA and CST production [62]. 

### 4.2. CST and Hypoxic-Inflammatory State

Physiological hypoxia and local inflammation are needed at the beginning of pregnancy to support the development of the placenta and ensure successful implantation. In PE, this state is prolonged, resulting in systemic inflammation and excessive ROS generation, which translates into increased oxidative stress. 

The anti-inflammatory role of CST is well established [63,64,65]. In vitro studies have found that the levels of pro-inflammatory interleukin 6 (IL-6), IL-1β, and TNF-α, known to be M1 macrophage markers, as well as their expression level, were significantly reduced after CST treatment [65]. Another study on human pulmonary artery endothelial cells (HPAECs) found that CST attenuates endothelial inflammation, partly by inhibiting TLR-4-p38 signaling activation. Toll-like receptor 4 (TLR4) protein expression was found to be higher in spontaneously hypertensive rats (SHR). After the administration of anti-TLR4 antibodies, the SHR demonstrated a reduced TLR4 protein level, mean arterial pressure, and pro-inflammatory IL-6 serum level in comparison to control rats. TLR4 is also an important activator of immune cells and modulates the materno–fetal interference. Changes in its expression and signaling can result in pregnancy disturbances [66]. In fact, a study of maternal excessive inflammatory responses in PE women revealed significantly higher endogenous TLR4 expression and NF-κB activation in monocytes from patients with PE [67]. Thus, studies on CST-TLR4 dependence and its influence on PE pathogenesis are worth considering.

Deteriorating systemic inflammation is directly connected to the prolonged hypoxic (2–3% O_2_) conditions observed in the PE placenta. A key transcription factor mediating cellular responses to low oxygen tension is hypoxia-inducible factor-1 (HIF-1). In physiological hypoxia, HIF-1 subunit α levels are elevated at the beginning of pregnancy, but significantly decrease after 10–12 weeks of gestation, which correlates with the oxygen influx to the placental bed. In PE, the HIF-1α level remains abnormally elevated, arresting the trophoblast in the first stage and resulting in shallow invasion [68]. Recent studies suggest the utility of HIF-1α as a predictive marker for PE [69,70]. 

An in vitro study conducted on HTR-8/SVneo cells, representing a first-trimester trophoblast, showed that HIF-1α regulates PAI-1 expression on the mRNA level. It was found that HIF-1α-specific siRNA reduced the PAI-1 protein level, indicating the important role played by HIF-1α in the hypoxia-mediated stimulation of PAI-1 expression and protein level [71]. Through stimulating PAI-1, HIF-1α may also contribute to a decrease in the level of plasminogen activators (PA): urinary PA (uPA) and tissue PA (tPA) [72]. It was shown that cells cultured in hypoxic conditions exhibited significant decreases in the levels of soluble uPA and tPA. Additionally, an imbalance was noted between high PAI-1 level and low uPA and tPA levels, which may have been part of the PE molecular mechanisms [73]. 

Both PAs are serine proteases catalyzing the conversion of plasminogen to plasmin. As mentioned before, plasmin is one of the enzymes implicated in the proteolytic cleavage of CgA to CST. Therefore, in a preeclamptic, hypoxic environment, changes in the levels of the two PAs may influence plasmin generation and subsequent CST formation. 

Furthermore, a study with fibronectin-coated pharmacologically active microcarriers (FN-PAM) examined whether the incorporation of CST into FN-PAM and its release affects the survival of mesenchymal stem cells (MSCs) under hypoxia-generated stress conditions. It was found that this controlled release of CST limited hypoxic MSC death and enhanced cell survival in the post-hypoxic environment [74]. Hence, it appears that effective CST delivery may be a potent approach for repairing the damage caused by prolonged hypoxia. 

### 4.3. CST and Apoptosis

Despite apoptosis being a crucial process associated with trophoblast formation and maternal spiral artery remodeling, it can disrupt trophoblast formation and vascularization in the excess inflammation and hypoxia characteristic of PE [75,76]. It was recently shown on two trophoblastic cell lines (HTR-8/SVneo and BeWO) that CST protein level negatively correlates with apoptotic ratio [62]. Furthermore, previous experiments indicate that CST can attenuate apoptosis by at least two different signaling pathways. An in vitro study conducted on cardiomyocytes found CST to activate the type 2 muscarinic acetylcholine receptor (MR2), which in turn activates ERK1/2 and PI3K/ Akt, which inhibits endoplasmic reticulum stress-induced cell death [77]. It has also been shown that MR2 is present in the endothelium of the coronary vasculature, where their activation is related to vasodilation [78]. MR2 are known to be expressed in the human placenta and, although their exact role in this tissue is not fully understood, some studies link MR2 receptors with PE [79,80]. At the same time, an in vitro study using matrigel assays with human coronary artery endothelial cells (HCAEC) confirmed the protective effect of CST against programmed cell death, and also emphasized that CST mediated capillary-like tube formation, confirming its positive influence on angiogenesis. 

Additionally, another in vitro study confirming the anti-apoptotic function of CST found it can regulate the PI3K/Akt signaling pathway by activating the beta 2 adrenergic (β2) receptor [81]. Although the activation of β2 receptors typically results in vasodilatation [82], it can induce an observable reduction in β2-mediated responses in hypertension and lower the number of functional β2-adrenoceptors in PE [83,84]. Thus, the protective action of CST acting through those receptors can be abolished in PE. Recent in vitro and clinical studies indicate that the CST protein level is diminished both in the placentas of PE patients and trophoblastic cell lines [30,62]. Therefore, further experiments are necessary to determine the relationship between apoptosis and CST, and the related signaling pathways that are disturbed in PE. 

### 4.4. CST in Signaling Pathways of PE

Trophoblast cell proliferation, migration, and fusion are highly controlled by numerous factors, including cytokines, growth factors, and hormones. Each signal generated by those factors initiates linear or branched signaling cascades, leading to the expression of effector molecules influencing the outcome of pregnancy. Several cell signaling pathways believed to play a role in trophoblast formation have been studied, together with the possible crosstalk between them. Some of these pathways influencing the molecular basis of pregnancy are believed to interact with CST (Figure 3).

#### 4.4.1. Mitogen-Activated Protein Kinases (MAPKs) Signaling Pathway

In the MAPK signaling pathway, receptor tyrosine kinases (RTKs) are activated by ligand–receptor interaction. This results in the phosphorylation of cytoplasmic domain tyrosine residues and the subsequent attraction of signaling molecules such as Grb-2 (growth factor receptor-bound protein 2) or SHP2, i.e., Src homology 2 (SH2)-domain containing tyrosine phosphatases. Next, Ras and Raf GTPases are activated, initiating a cascade of signals resulting in the final activation of MAPK families: ERKs, JNKs (c-Jun N-terminal kinases), and p38 MAPKs [85].

During pregnancy, MAPK signaling is involved in trophoblast invasion, fusion, and differentiation [85]. In PE, impaired trophoblast invasion is partly modulated by abnormal MAPK1/ERK2 signaling [86]; in addition, MAP/ERK molecules are associated with pathways related to PE endothelial dysfunction and immunological alterations [87]. 

According to Feng Liao et al., CST can reduce endoplasmic reticulum (ER)-mediated stress and increase cell survival through the activation of the ERK1/2 pathway [77]. The same effect is evoked by CST via the activation of the PI3K/Akt pathway. Although these studies were conducted on cardiomyocytes, it is possible that similar effects can be obtained in placental cells. Furthermore, an in vitro study conducted on placentas from women with and without PE revealed that the c-Src-MAPK (ERK1/2, p38, and JNK)-NFKB signal transduction pathway is downregulated in PE. C-Src is a Src family protein tyrosine kinase (SFK) that plays important roles in trophoblast formation and syncythialization [88]. Another in vitro study conducted on human aortic smooth muscle cells (HASMCs) suggested that CST can significantly increase c-Src and PI3K protein expression, as well as ERK1/2 phosphorylation [89]. There is a need for further comprehensive studies on CST signaling through the c-Src-MAPK pathway, as the results may shed valuable light on the placental dysfunctions observed in PE. 

#### 4.4.2. Phosphoinositide 3-Kinase (PI3K)/Akt Signaling Pathway

The PI3K/AKT pathway begins with the activation of RTKs or G-protein-coupled receptors (GPCRs), followed by the recruitment of p85 and p110 subunits of PI3K to the membrane. Subsequently, phosphatidylinositol-4 and 5-bis-phosphate (PIP2) are phosphorylated and converted to PIP3. Increased PIP3 level activates AKT by phosphoinositide-dependent kinase-1 (PDK1), which is followed by the activation of downstream targets, such as a mammalian target of rapamycin (mTOR) [85]. The PI3K/Akt/mTOR signaling pathway is known to regulate, among others, decidualization, implantation, cell growth, proliferation, migration, and survival [90,91,92]. 

In addition to its anti-apoptotic effect, acting through the ERK1/2 pathway, CST can also inhibit endoplasmic reticulum stress-induced cell apoptosis with the PI3K/Akt pathway [80]. In addition, CST can promote NO release from endothelial cells by PI3K/Akt-dependent nitric oxide synthase (eNOS) phosphorylation; this may, in part, be responsible for its protective, anti-hypertensive action [52]. It has been recently shown that inhibiting the PI3K/Akt/mTOR axis in HTR8/SVneo trophoblast representing cells can contribute to PE development [91]; also, activation of this pathway increased proliferation and migration in the trophoblast cells, and suppressed oxidative stress, suggesting that it has beneficial effects. CST was shown to enhance phospho-Akt signals by stimulating PDK-1 and mTORC2 (mTOR Complex 2; rapamycin-insensitive complex) activities [92]. In an in vivo study conducted on wild-type (WT) and CST-KO mice, CST stimulated glycogenesis, reduced glycogenolysis and glucose production, and enhanced downstream insulin signaling. Considering the possible protective action of CST, the fact that women with PE can exhibit insulin resistance [93], and that the regulation of glycogenesis and glycogenolysis pathways in PE are not fully understood [94], further studies on the influence of CST on PE development are merited. 

In contrast, some research suggests that PI3K/Akt activation may be connected to PE progression [95]. However, these findings may be influenced by the fact that the effect of PI3K/Akt pathway activation or inhibition on pregnancy outcome depends partially on the stage of pregnancy. For instance, it was shown that at the beginning of pregnancy, in physiological hypoxia, HIF-1α may act through the PI3K/Akt pathway promoting cell survival, while during prolonged ischemia, the same relationship may induce cell death [96]. Also, HIF-1α was found to have a strong relationship with the mTOR signaling pathway, which influenced embryonic organ development [97]. According to Kimura et al., HIF-1α negatively regulates the mTOR axis and suppresses salivary gland development under 1% O_2_. As such, there is a need for deeper research into the possible relationship between prolonged hypoxia observed in PE, CST, and mTOR signaling. 

#### 4.4.3. JAK-STAT Signaling Pathway

In the Janus kinase/signal transducer and activator of transcription (JAK/STAT) axis, after receptor–ligand binding, the receptor dimerizes and phosphorylates its own JAKs and cytoplasmic domains. Consequently, those domains can bind phosphotyrosine-binding (PTB) domains or proteins having SH2-like STATs and phosphorylate them. The activated STATs dissociate and translocate into the nucleus, where they regulate transcription by binding to specific promoters of the targeted genes [88]. The JAK/STAT axis is involved in the regulation of embryogenesis, trophoblastic cell proliferation, and invasion. 

An in vivo study on mice revealed that STAT3 is essential for the early development of mouse embryos [98]. Also, an in vitro study with HTR8/SVneo cells found STAT1 and STAT3 phosphorylation play a critical role in endothelial growth factor (EGF)-mediated placental cell invasion. This process has also been associated with possible cross-talk between the ERK1/2 and JAK/STAT pathways [99]. CST activates both ERK1/2 and STAT3 phosphorylation. CST-mediated STAT3 activation was shown to exhibit an anti-inflammatory effect and influence cell proliferation and migration [64]. Cirri et al. report that the c-Src tyrosine kinase is involved in STAT3 activation [100], and therefore, its activity could be indirectly mediated by CST, as CST increases c-Src protein expression. It has also been found that STAT3 with NF-κB can cooperatively regulate a number of target genes. For instance, the activation of both factors together can prevent apoptosis in cancer cells [101]. 

#### 4.4.4. NF-κB Signaling Pathway

The transcription factor NF-κB is not only engaged in inflammation, but also in apoptosis and immunity. In its inactive form, NF-κB is located in the cytoplasm, where it is bound to its inhibitor IkB (IkBα or IkBβ). After its activation, NF-κB forms a complex with subunits (e.g., p65:p50) and translocates into the nucleus, where it acts as a transcription factor. Generally, there are three main NF-κB activation pathways: canonical, non-canonical, and atypical. 

One of the receptors involved in NF-κB activation is the plasma membrane TLR4. As mentioned before, women with PE demonstrate higher TLR4 expression and NF-κB activation [66]. Furthermore, an in vivo study conducted on pregnant rats found a single treatment with lipopolysaccharide (LPS; a TLR4 agonist) to result in elevated expression of both TLR4 and NF-κB in the placenta, which was accompanied by FGR and adverse pregnancy outcomes [102]. Therefore, it is possible that TLR4 and NF-κB levels are elevated not only as a consequence of PE, but also can contribute to its development. A recent study found most of the commonly studied NF-κB activation pathways to be downregulated in PE, and that the pathway responsible for its activation in trophoblastic cells may involve the tumor protein p53/ribosomal protein S6 kinase alpha-1 (p53/RSK1) complex [103]. 

Although, generally, p53 and NF-κB signaling contradict each other, there are a few examples of them working together. For instance, p53 has been found to activate NF-κB, which correlated with the ability of p53 to induce apoptosis [104]. In addition, a unique NF-κB/p53 interaction was seen to drive the expression of chemokines and cytokines, e.g., IL-6, in macrophages [105]. Interestingly, an in vitro study conducted on dendritic cells revealed that p53 overexpression reduced LPS/ TLR4-mediated NF-κB-gene-dependent transcription [106]; however, it was proved that p53 expression is dependent on oxygen level [107]. For instance, HIF-1 downregulates p53 under mild hypoxia, but mediates its activation in severe or prolonged hypoxia [108]. Further, p53 upregulation and apoptosis in hypoxia were shown to be specific to cytotrophoblastic cells, whereas hypoxic activation was correlated with p53 downregulation in syncytiotrophoblast HIF-1, thereby limiting p53-dependent cell death [109,110]. Therefore, it appears that the prolonged hypoxia observed in the PE placenta may alter p53 downstream targeting, altering TLR4/p53/NF-κB-gene-dependent transcription. Indeed, all of the proteins mentioned above are overexpressed in PE, and TLR4 was reported to increase p53 activity [111]. Activated p53 can further cooperate with RSK1, and the newly formed complex may activate NF-κB in the nucleus (Figure 4). 

As such, further studies on CST-dependent molecular signaling pathways may shed greater light on its influence on embryogenesis and placental development, and the possible cross-talks between the axes involved in those processes. 

## 5. Conclusions

CgA-derived CST is produced and released by the placental cells. It can exert multiple effects influencing placental vascular and cellular development, such as angiogenesis, migration, proliferation, apoptosis, and vascular tension. The molecular pathways associated with CST are well documented, and supported by numerous in vitro, in vivo, and clinical studies. Many of the CST-related processes are disturbed in PE, in which the level of CST protein is known to be diminished compared to healthy pregnancies; similar tendencies are observed in patients with hypertension: one of the key characteristics of PE. Hence, there exists strong evidence that the CST protein has a possible influence on processes associated with the development of PE. Further studies are needed to fully understand the framework of CST molecular signaling in pregnancy. Additionally, our knowledge of the effect of CST on the symptoms of PE remains incomplete. However, since such studies raise ethical concerns, they should be substituted with in vitro and in vivo experiments reflecting the PE environment. 

## Figures and Tables

**Figure 1 ijms-25-02461-f001:**
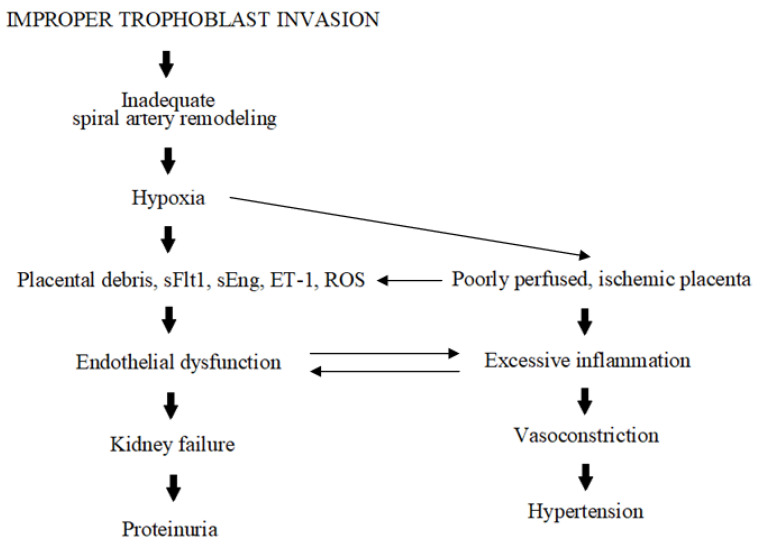
Schematic presentation of PE pathogenesis.

**Figure 2 ijms-25-02461-f002:**
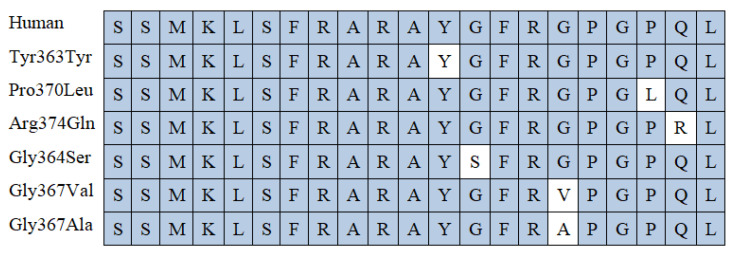
Comparison of catestatin peptide region in human genetic variants and rat. White color is reserved for changes in the amino acid sequence.

**Figure 3 ijms-25-02461-f003:**
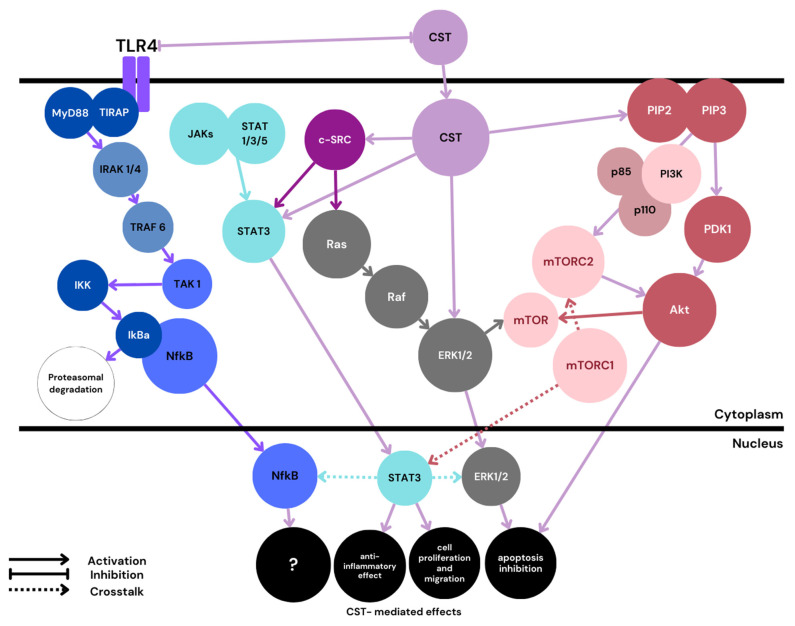
**Proposed schematic.** PE-related CST signaling pathways. MyD88—myeloid differentiation factor 88; TIRAP—toll-interleukin-1 receptor containing adaptor protein; IRAK—IL-1 receptor-associated kinase; TRAF—TNF receptor-associated factor; TAK—tat-associated kinase; IkB—inhibitor of Nf-κB; IKK—IkB kinase; mTOR—mammalian target of rapamycin; mTOR1/2—mTOR complex 1/2; PIP2—phosphatidylinositol 4,5-bisphosphate; PIP3—phosphatidylinositol 3,4,5-trisphosphate; PDK1—phosphoinositide-dependent kinase-1. Color blue is reserved for the Nf-κB classical pathway mediated through TRL4. CST signaling is marked with a light violet color.

**Figure 4 ijms-25-02461-f004:**
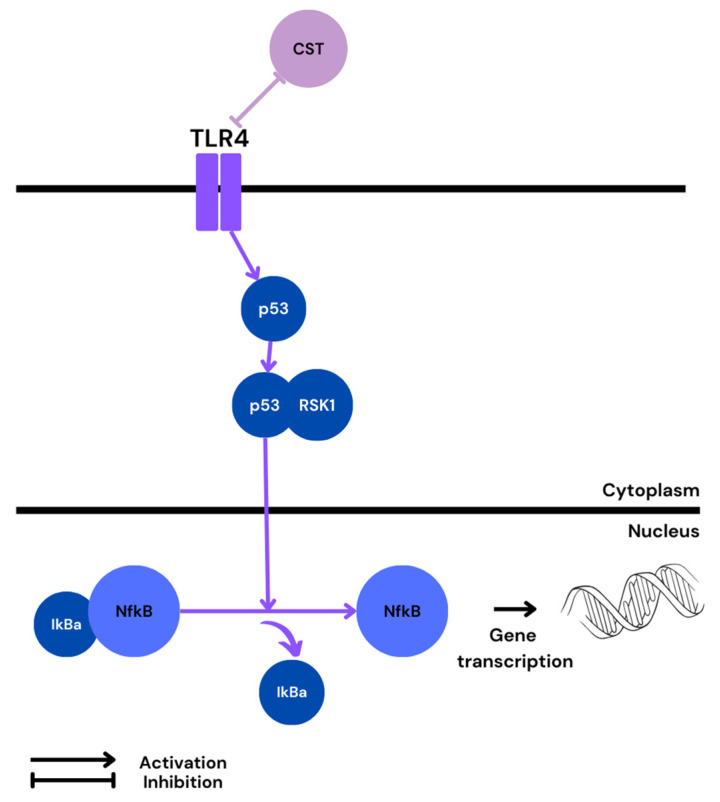
Possible pathway of NF-κB activation in PE and predicted CST influence.

**Table 1 ijms-25-02461-t001:** Diagnostic criteria of preeclampsia according to the American College of Obstetricians and Gynecologists (ACOG) and The International Society for the Study of Hypertension in Pregnancy (ISSHP) [4,5].

Diagnostic Criteria	ACOG	ISHHP
Blood pressure	SBP ≥ 140 mmHg or DBP ≥ 90 mmHg, on 2 separate occasions, at least 4 h apart, after 20 wk of gestation in previously normotensive woman SBP ≥ 160 mmHg or DBP ≥ 110 mmHg, on different, short periods of time (min)	SBP ≥ 140 mmHg or DBP ≥ 90 mmHg, on 2 separate occasions SBP ≥ 160 mmHg or DBP ≥ 110 mmHg, on 2 separate occasions, 15 min apart
Proteinuria	Protein ≥ 300 mg in 24 h collective of urine OR Protein/creatinine ratio ≥ 0.3 mg/dL OR Dipstic test ‘+2’	Dipstic test ‘1+’, ≥30 mg/dL AND (if previous positive) protein/creatinine ratio ≥ 0.3 mg/dL
Hematological complications	Thrombocytopenia-number of thrombocytes below 100,000 × 10^9^/L	Thrombocytopenia-number of thrombocytes below 150,000/µL DIC, hemolysis
Renal damage	Kidney damage: plasma creatinine concentration above 1.1 mg/dL or double the plasma creatinine concentration with lack of other kidney diseases Kidney malfunction: twice elevated concentration of liver transaminases in blood	Acute kidney injury (creatinin ≥ 90 µmol/L; 1 mg/dL)
Uteroplacental dysfunctions	-	FGR, abnormal results from the Doppler examination of the umbilical artery, miscarriage
Neurological complications	New episode of drug-resistant headache, impossible to diagnose or showing no other symptoms	Eclampsia, altered mental state, blindness, stroke, heavy headaches, clonus, persistent escotoma
Other maternal organ dysfunctions	Pulmonary oedema	Liver complications (elevated transaminases e.g., ALT OR AST > 40 IU/L) with or without pain in the lower right abdomen or upper abdomen)

ALT—alanine transaminase, AST—aspartate transaminase, DIC—disseminate intravascular coagulation, and FGR—fetal growth restriction.

## Data Availability

Data are contained within the article.

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
