# Peer review of "The Role of Catestatin in Preeclampsia"

_ijms, 2024, doi:10.3390/ijms25052461_

Round 1
Reviewer 1 Report
Comments and Suggestions for Authors
This is an interesting review, which is focussed on catestatin, peptide derived from chromogranin A, which is produced and released by the placental cells. The authors show that catestatin can exert multiple effects influencing placental vascular and cellular development, such as angiogenesis, migration, proliferation, apoptosis and vascular tension. Catestatin molecular pathways are well documented in this study pointing out also on contradictory data.
Major point
1. Chromogranin A is found in neuroendocrine organs and is used as a marker of the sympatho-adrenal neuroendocrine activity. Moreover, proteolytic processing of chromogranin A gives rise to an array of biologically active peptides such as, vasostatin, WE14, catestatin, and serpinin, which have diverse roles in regulating cardiovascular functions and metabolism, as well as inflammation. Therefore, discuss in the MS that under “in vivo” conditions effects of chromogranin A can be modifies or even counterbalanced by activated sympatho-adrenal neuroendocrine system or active peptides derived from chromogranin A.
2. Intricate tissue-specific role of ChgA-derived peptide activity in preclinical rodent models of metabolic syndrome reveals complex effects on carbohydrate and lipid metabolism. Catestatin can play a role in metabolic syndrome such as obesity, insulin resistance, and diabetes mellitus. Plese, discuss if these effects can modify effects of catestatin described in the MS?
Minor points
p. 303 cardiomiocytes should be cardiomyocytes.
p. 328 can epithelial-mesenchymal transition indicate only positive effects? Risky for cancer development.
p. 414 … many CST related processes is disturbed in PE… please correct
Comments on the Quality of English LanguageMinor editing of English language required.
Author Response
REVIEWER 1:
The authors want to thank all of the Reviewers for the time spent on the article and for their valuable comments. The authors tried to do their best to address each comment and to apply solutions proposed by the Reviewers. To make sure that the manuscript is easier to read and do not include any other language errors, the whole paper was proof- read by the English native speaker.
Major point
- Chromogranin A is found in neuroendocrine organs and is used as a marker of the sympatho-adrenal neuroendocrine activity. Moreover, proteolytic processing of chromogranin A gives rise to an array of biologically active peptides such as, vasostatin, WE14, catestatin, and serpinin, which have diverse roles in regulating cardiovascular functions and metabolism, as well as inflammation. Therefore, discuss in the MS that under “in vivo” conditions effects of chromogranin A can be modifies or even counterbalanced by activated sympatho-adrenal neuroendocrine system or active peptides derived from chromogranin A.
A: The Authors hope, that addition of the following paragraph will answer the first two questions of the Reviewer:
‘2. Pregnancy, the sympathoadrenal neuroendocrine system and its association with preeclampsia
The extensive systemic and local adaptations of maternal cardiovascular system characterize each complicated and non-complicated gestation. These are enable to support the growing metabolic demands of the fetus as well as to allow for the delivery of oxygen and nutrients to the developing placenta. Among the systemic, hemodynamic adaptations the most spectacular ones are as follows: (1) an increased cardiac output presenting the sharpest rise at the beginning of the gestation and the maximum peak at the early third trimester, and (2) the elevation of maternal blood volume being accompanied by gradually reduction in peripheral resistance of maternal vessels. The latter one is possible thanks to the systemic vasodilatation of maternal vessels appearing before the end of placentation and completion of uteroplacental circulation, i.e. before the week 5 of gestation. Interestingly, some of the maternal adaptations of vascular system begin even before gestation i.e. during the luteal phase of a menstrual cycle. At that time, the elevation of progesterone level is accompanied by an increase of density of maternal functional β2 adrenergic receptors, although the general density of β2 adrenergic receptors is unaltered or even depleted in gestation, in comparison to non-pregnant women (6,7). These receptors mediate the catecholamine, especially norepinephrine and epinephrine, response and play a significant role in the fall down of the maternal vascular resistance. Indeed, the activation of β-adrenergic receptors causes vasodilatation of vascular bed and additionally leads to the relaxation of uterine muscles (8). Moreover, the study on transgenic mice with overexpression of β2 adrenergic receptors demonstrates that these receptors upregulate the cardiac output and heart rate (9). The effect of action of adrenergic receptors on the cardiovascular system was also observed in humans (10). Therefore, it is not surprise that the sympathoadrenal neuroendocrine system is changed in pregnant, in comparison to non-pregnant women. Interestingly, the uterine levels of norepinephrine, as well as dopamine, are significantly lower in normal pregnant patients when compared to non-pregnant women (11), as well as in comparison to hypertensive pregnant women (12). Indeed, the preeclamptic patients present the higher levels of studied catecholamines in uterine tissues with the greater elevation in the placental bed and a higher level of norepinephirne in plasma in comparison to non-hypertensive expectant women (11). Additionally, although the epinephrine level does not vary significantly between preeclamptic and normotensive pregnant women (12), the epinephrine/norepinephrine ratio is significantly elevated in women whose pregnancy is complicated by PE (11). It is possible that the higher level of catecholamines in preeclamptic women is an effort of maternal organism to stimulate the β2 adrenergic receptors. Indeed, the affinity state of β2 receptors in maternal tissue of normotensive women is much higher than in the preeclamptic patients (6). This depletion in density of functional β2 receptors in preeclampsia seems to be related to the reduction of total number of β2 adrenergic receptors in maternal tissues than the loss of their function (6). This low affinity state of adrenergic receptors may explain why the glucose metabolism is dysregulated in preeclamptic women (13). The β2 adrenergic receptors are involved in the glucose uptake both in vitro myotubes and the skeletal muscle of healthy subject (14). The ablation of β2 adrenergic receptors in mice results in hyperglycemia, but the supplementation of obese mice or the rat model for human type 2 diabetes mellitus by the β2 adrenergic agonists improve the glucose tolerance in these rodents (14-16). The state of reduction in the density of functional β2 receptors in preeclamptic women may have an additional consequence in maternal response to the other stimulators of these receptors, including the CgA and its derivative peptides such as CST (17). Especially, that the depletion of CST in blood and placental bed of preeclamptic women observed in numerous studies (18-20), seems to deepen this pathological state. It is possible that CST cooperates with the adrenergic receptors in the regulation of generation of nitric oxide (NO) being known from its vasodilatory effect on the maternal vessel (21) Moreover, an in vitro study demonstrates that CST action through the β2 adrenergic receptors preserves the apoptotic process of cardiomyocytes in the response to oxidative stress (22). CST is also perceived as an endocrine/paracrine cardiac modulator; the inotropic effect of CST is mediated mainly by the β2 adrenergic receptors (23). Interestingly, CST action by adrenergic system also influences the fat deposit as well as induces the lipolysis process and fatty acid oxidation. Fat cell function is regulated by the adrenergic system including the β2 adrenergic receptors which activation stimulate the lipolysis process. The in vitro studies, indicate that high level of catecholamines observed in obese patients, as well as in preeclamptic women (12), lead to the β adrenergic and leptin receptors desensitization and to disturbances in the secretion of leptin. These processes are normalized by the chronic CST treatment (24).’
- Intricate tissue-specific role of ChgA-derived peptide activity in preclinical rodent models of metabolic syndrome reveals complex effects on carbohydrate and lipid metabolism. Catestatin can play a role in metabolic syndrome such as obesity, insulin resistance, and diabetes mellitus. Plese, discuss ifthese effects can modify effects of catestatin described in the MS?
A: Question addressed in the previous one.
Minor points
- 303 cardiomiocytes should be cardiomyocytes.
A: The error was corrected.
- 328 can epithelial-mesenchymal transition indicate only positive effects? Risky for cancer development.
A: The authors agree that EMT not always exert positive effects, yet, in the cited article observed changes were positive. However, to avoid contradictory reception of the text the mentioned fragment was removed.
- 414 … many CST related processes is disturbed in PE… please correct
A: The error was corrected. Additionally, the whole text was corrected by the native English speaker.

Reviewer 2 Report
Comments and Suggestions for Authors
In this review, the authors discuss the effect that catestatin (CST) has in multiple disorders and molecular pathways that are relevant to preeclampsia (PE). The authors collect multiple studies that provide evidence suggesting that CST plays a role in the pathophysiology of PE and also present gaps in the current understanding of the influence CST has in PE.
The connection between CST and estrogen receptors is unclear. It seems like a reach to suggest that CST mediates in actions through the estrogen receptor because they cause similar vasodilatory downstream actions.
In figure 1, kidney failure is a more extreme condition than kidney damage. While PE may cause kidney damage, kidney failure is a less likely outcome and should not be included in a general schematic of the pathophysiology.
The relationship between the genetic polymorphism and genome wide association studies to PE is not clear in their specific paragraphs.
Figure 3 is difficult to read, a higher resolution or larger image would be helpful.
It is not clear how CST affects angiogenesis. If there are specific studies demonstrating a pro-angiogenic effect of CST, they should be discussed before the conclusion sentence: “It can exert multiple effects influencing placental vascular and cellular development, like: angiogenesis, migration, proliferation, apoptosis and vascular tension.”
A discussion of the difference in gestational hypertension and PE should be added as it seems many of the studies included demonstrate a link between CST and hypertension, but not specifically PE. Also any differences in CST between early onset PE and late onset PE might provide some insight into studies that show an increase in CST and studies that show a decrease in CST in PE pregnancies.
Comments on the Quality of English LanguageThere are a few sections throughout the paper that require some English Language editing.
Some sentences are awkward and could use a rewrite for clarification.
Author Response
The authors want to thank all of the Reviewers for the time spent on the article and for their valuable comments. The authors tried to do their best to address each comment and to apply solutions proposed by the Reviewers. To make sure that the manuscript is easier to read and do not include any other language errors, the whole paper was proof- read by the English native speaker.
REVIEWER 2:
- The connection between CST and estrogen receptors is unclear. It seems like a reach to suggest that CST mediates in actions through the estrogen receptor because they cause similar vasodilatory downstream actions.
Answer: Mentioned assumption was not only based on the similarity of vasodilatory actions, but also on the fact that ER are numerous in female organism and that clinical studies prove, that CST vasodilatory effect is more evident in female patients comparing to man. Further, It was shown that β-adrenergic agonists (like CST) can modulate estrogen receptor (ER)-regulated gene expression. Such cross- talk sugessts possibility of usage of β-adrenergic agonists in hypertension treatment, considering possible gender-related differences in cardiovascular regulation*.
* Walters MR, Sharma R. Cross-talk between beta-adrenergic stimulation and estrogen receptors: isoproterenol inhibits 17beta-estradiol-induced gene transcription in A7r5 cells. J Cardiovasc Pharmacol. 2003 Aug;42(2):266-74. doi: 10.1097/00005344-200308000-00017. PMID: 12883332.
- In figure 1, kidney failure is a more extreme condition than kidney damage. While PE may cause kidney damage, kidney failure is a less likely outcome and should not be included in a general schematic of the pathophysiology.
A: The authors agree and apologize for this mistake, which is a translation error. The phrases were changed as follows: the kidney failure was changed to kidney damage, renal failure to renal damage and acute renal failure to acute kidney injury (all in Table 1).
- The relationship between the genetic polymorphism and genome wide association studies to PE is not clear in their specific paragraphs.
The whole manuscript was corrected by the English native speaker and the authors hope that above will make the article easier to read. Also, the following sentence has been removed, as it was consider irrelevant to the described topic:
Lines 165-166: “Interestingly, another genome-wide association study one KLKB1 genetic variant to be associated with a high level of ET-1 protein surrogate.”
- Figure 3 is difficult to read, a higher resolution or larger image would be helpful.
A: Figure 3 was enlarged and replaced in the manuscript.
- It is not clear how CST affects angiogenesis. If there are specific studies demonstrating a pro-angiogenic effect of CST, they should be discussed before the conclusion sentence: “It can exert multiple effects influencing placental vascular and cellular development, like: angiogenesis, migration, proliferation, apoptosis and vascular tension.”
A: The authors agree that this issue was too superficially, and only one sentence before mentioned angiogenic properties of catestatin (Lines 69-71: ‘CST is also a potent inhibitor of neuropeptide Y, adenosine triphosphate and CgA secretion, and has been found to manifest angiogenic and vasorelaxant activities..’). Thus, a following sentence was added:
Lines 251-254: ‘In the same time, an in vitro study using matrigel assays with human coronary artery endothelial cells (HCAEC) confirmed protective effect of CST against programmed cell death, and also emphasized that CST mediated capillary like tube formation, confirming its positive influence on angiogenesis.’
- A discussion of the difference in gestational hypertension and PE should be added as it seems many of the studies included demonstrate a link between CST and hypertension, but not specifically PE. Also any differences in CST between early onset PE and late onset PE might provide some insight into studies that show an increase in CST and studies that show a decrease in CST in PE pregnancies.
A: The whole new paragraph entitled: ‘2. Pregnancy, the sympathoadrenal neuroendocrine system and its association with preeclampsia’was added.
Further, the key differences (including pathogenetic background) between GH and PE have been included in the following sentences (Lines 41-45):
‘Although among HDP in both, PE and GH, diagnosis criteria encompass recognition of hypertension during pregnancy in the previously normotensive women, the diseases are two distinct entities, as in the PE a damage of one of the end- organ must be confirmed. Furthermore, epidemiologic, pathogenetic and hemodynamic characteristics are different for both of them.’
Regarding CST and PE: up till now only 4 articles present studies on CST in PE*. None of them points statistically significant differences between CST level in early and late PE groups. However, as the authors suggest in the manuscript (Lines 176-178), the discrepancy between CST level in PE (lower or higher) may result from the differences in applied inclusion criteria and further, from the difference in CST protein production observed during different stages of pregnancy; alternatively, its production may be influenced by maternal compensatory mechanisms, or the CST level in the serum of PE mothers may be derived from other origins than the placenta, resulting in a different systemic protein level to that present in the placenta.
* Bralewska M, Biesiada L, Grzesiak M, et al.(2021). Chromogranin A demonstrates higher expression in preeclamptic placentas than in normal pregnancy. BMC Pregnancy Childbirth. 21(1):1-10. doi:10.1186/S12884-021-04139-Z/FIGURES/3
* Özalp M, Yaman H, Demir Ö, Garip SA, Aran T, OsmanaÄŸaoÄŸlu MA(2021). The role of maternal serum catestatin in the evaluation of preeclampsia and fetal cardiac functions. Turkish J Obstet Gynecol. 18(4):272. doi:10.4274/TJOD.GALENOS.2021.34946
* Tüten N, Güralp O, Gök K, et al.(2022). Serum catestatin level is increased in women with preeclampsia. J Obstet Gynaecol. 42(1):55-60. doi:10.1080/01443615.2021.1873922
* Palmrich P, Schirwani-Hartl N, Haberl C, Haslinger P, Heinzl F, Zeisler H, Binder J. Catestatin-A Potential New Therapeutic Target for Women with Preeclampsia? An Analysis of Maternal Serum Catestatin Levels in Preeclamptic Pregnancies. J Clin Med. 2023 Sep 12;12(18):5931. doi: 10.3390/jcm12185931. PMID: 37762872; PMCID: PMC10531844.
- There are a few sections throughout the paper that require some English Language editing. Some sentences are awkward and could use a rewrite for clarification.
A: As mentioned earlier in the answers, the whole manuscript was corrected by the English native speaker and the authors hope that above will make the article easier to read.

Reviewer 3 Report
Comments and Suggestions for Authors
The work attempted to clarify the role of catestatin in PE, however, the links between CST and PE is described poorly, while the conclusions are made by analogy with other hypertensive conditions. I understand that CST was implicated in PE development quite recently and information on this subject is scarce, but in such case the summarizing its role in PE is premature. Besides, the article is poorly structured and requires an extensive improvement before considering it for publication. The points are below:
1. Abstract, lines 13-15: I did not find the comprehensive description of in vitro, in vivo and clinical studies revealing the role of CST in PE molecular processes in the main text.
2. I would recommend to strictly divide the role and functions of CST in hypertensive disorders in general and in PE by adding new subsections or restructuring existing ones. For example, Lines 130-148, 174-194, 201-210 can be moved to other sections.
3. Section 3.4: the conclusions on involvement of described signaling pathways to PE are not supported by experimental data, probably due to their absence. Accordingly, the signaling cascades presented on Fig. 3 are quite speculative. This concerns, for example, MAPK pathway. Moreover, in some places the literature data are cited incorrectly. For example, reference [73] is the work performed on cultured trophoblast cells, not on PE and healthy women placentas (lines 304-307). I would recommend to rename the Figure 3 using the term “proposed” or something like that.
4. The text contains a lot of orthographic errors. For example:
Table : pulmonary oedma,
Line 72: angogenic.
Authors should carefully read the text, sentence by sentence, to correct all the errors.
Author Response
The authors want to thank all of the Reviewers for the time spent on the article and for their valuable comments. The authors tried to do their best to address each comment and to apply solutions proposed by the Reviewers. To make sure that the manuscript is easier to read and do not include any other language errors, the whole paper was proof- read by the English native speaker.
REVIEWER 3:
The work attempted to clarify the role of catestatin in PE, however, the links between CST and PE is described poorly, while the conclusions are made by analogy with other hypertensive conditions. I understand that CST was implicated in PE development quite recently and information on this subject is scarce, but in such case the summarizing its role in PE is premature. Besides, the article is poorly structured and requires an extensive improvement before considering it for publication. The points are below:
- Abstract, lines 13-15: I did not find the comprehensive description of in vitro, in vivo and clinical studies revealing the role of CST in PE molecular processes in the main text.
A: The authors agree, that the sentence was misleading, therefore the world ‘proposed’ was added as follows:
‘Therefore, the aim of the present paper is to review the most recent and most relevant in vitro, in vivo and clinical studies to provide an overview of the proposed impact of CST on the molecular processes of PE and to consider the possibilities for future experiments in this area.‘
Furthermore, to emphasize, that the presented manuscript is aimed to stress a link between the existing in the literature functions of CST and key characteristics of PE pathogenesis, using limited data from PE-based studies and additional relevant literature, the following sentence was added (Lines 58-62):
‘Despite the fact, that CgA itself was found in trophoblast in 1996, up till now it is not entirely clear what role in pregnancy and its complications, like PE, it may act. Thus, presented review attempts to fulfill existing gap and to present some new insights in the possible involvement of CgA- derived CST in PE pathomechanism.’
The authors hope that above will make the purpose of the article more evident, and also that it will clarify that presented in vitro, in vivo and clinical outcomes are not only derived from PE- based studies, but also from different topics, with the use of which the analogy of CST actions was translated into the possible events observed in PE patients.
- I would recommend to strictly divide the role and functions of CST in hypertensive disorders in general and in PE by adding new subsections or restructuring existing ones. For example, Lines 130-148, 174-194, 201-210 can be moved to other sections.
A: The whole new paragraph entitled: ‘2. Pregnancy, the sympathoadrenal neuroendocrine system and its association with preeclampsia’ was added.
Further, the authors want to clarify that the aim of the mentioned paragraph (in which the first two line sets are placed) entitled ‘CST and blood pressure regulation’ was to describe most relevant studies regarding vasodilatory role of CST and its relationship with hypertension. Presented outcomes of chosen studies were translated into the possible analogy in PE. The lines 201-210 (currently lines: 208-214) are indeed placed in the different paragraph and in part concern blood pressure regulation. As the original sentences put too much emphasize on hypertension issue, the fragment has been changed to better match the paragraph title ‘CST and hypoxic-inflammatory state’:
‘Toll-like receptor 4 (TLR4) protein expression was found to be higher in spontaneously hypertensive rats (SHR). After administration of anti-TLR4 antibodies, the SHR demonstrated reduced TLR4 protein level, mean arterial pressure and pro- inflammatory IL-6 serum level in comparison to control rats.’
- Section 3.4: the conclusions on involvement of described signaling pathways to PE are not supported by experimental data, probably due to their absence. Accordingly, the signaling cascades presented on Fig. 3 are quite speculative. This concerns, for example, MAPK pathway. Moreover, in some places the literature data are cited incorrectly. For example, reference [73] is the work performed on cultured trophoblast cells, not on PE and healthy women placentas (lines 304-307). I would recommend to rename the Figure 3 using the term “proposed” or something like that.
A: Caption of the figure 3 has been changed to ‘Proposed schematic PE- related CST signaling pathways.’. Additionally, the authors apologize for the citation mistake, as one of the literature position had been omitted. A new position was added where appropriate (related to PE and healthy placentas)*.
* Irtegun S, Akcora-Yıldız D, Pektanc G, Karabulut C. Deregulation of c-Src tyrosine kinase and its downstream targets in pre-eclamptic placenta. J Obstet Gynaecol Res. 2017 Aug;43(8):1278-1284. doi: 10.1111/jog.13350. Epub 2017 May 19. PMID: 28544129.
- The text contains a lot of orthographic errors. For example:
Table : pulmonary oedma,
Line 72: angogenic.
Authors should carefully read the text, sentence by sentence, to correct all the errors.
A: Both errors were corrected and the whole manuscript was proof-read by the English native speaker. The authors hope that above will make the article easier to read.

Round 2
Reviewer 3 Report
Comments and Suggestions for Authors
None